# Seeing the Unseen: Physics-as-Representation for Generalizable Gaze Perception

**Yunfeng Xiao** [1 2]   **Xiaowei Bai** [2 3]   **Hao Su** [2 4]   **Hao He** [2 3]   **Liang Xie** [2 3]   **Erwei Yin** [2 3 *]

## Abstract

We introduce physics-as-representation, a learning paradigm that encodes physical structure and geometric laws into visual representations, enabling models to see the unseen—the underlying 3D geometry and motion dynamics not apparent in raw pixels. We instantiate this paradigm in gaze perception by proposing SG-Gaze, a framework that learns a Structurally and Geometrically Consistent Representation (SGR) through dual-branch adversarial learning. An analytical branch embeds appearance features onto a spherical manifold aligned with gaze geodesics, while a model-guided branch reconstructs the 3D eyeball with weak 2D edge supervision. We further introduce View-Consistent Regularization, which augments SGR learning with synthetic view perturbations and enforces rotation-equivariant consistency across gaze vectors and structural projections, eliminating the need for multi-view calibration or explicit 3D labels. Extensive experiments across 12 challenging cross-domain transfers demonstrate that SG-Gaze achieves state-of-the-art accuracy and strong generalization. Our work highlights that enforcing structural and geometric consistency with equivariant regularization serves as effective inductive biases for interpretable and generalizable representation learning—a step toward machines that perceive the world not only from pixels, but from physics.

## 1. Introduction

Eye gaze estimation has broad applications in human–computer interaction (Sharma & Abrol, 2013; Terzioğlu et al., 2020), driver monitoring (Yuan et al., 2022; Cheng et al., 2024), and immersive AR/VR systems (Burova

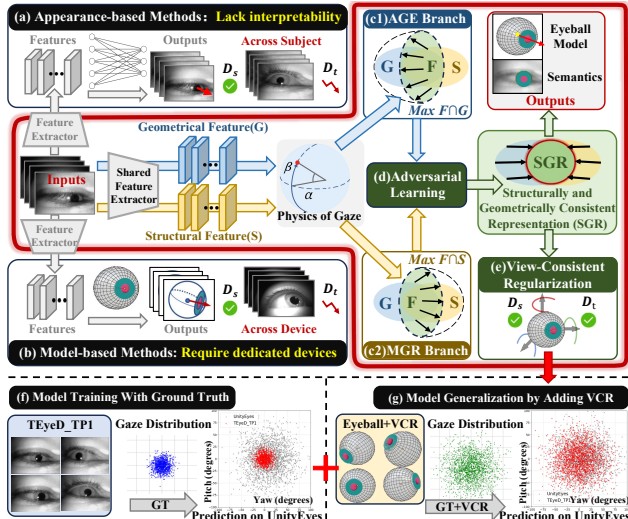

*Figure 1.* Overview of gaze estimation methods. (a) Appearance-based methods lack geometric constraints and interpretability. (b) Model-based methods require dedicated devices. (c) Our SG-Gaze unifies both methods by learning the most geometric (**G**) and the most structural (**S**) features grounded in gaze physics. (d) Adversarial learning fuses features into a unified Structurally and Geometrically Consistent Representation (SGR). (e-g) VCR enforces rotation-equivariant consistency to cover a larger field-of-view to mitigate domain gaps compared to GT supervision alone.

et al., 2020; Konrad et al., 2020). Beyond practical use, it serves as a representative case study for a broader challenge in machine learning: how to learn physically grounded and interpretable representations that generalize under distribution shifts. Accurate gaze estimation thus becomes a test of whether a model respects three physical priors: the spherical manifold of gaze directions, the anatomical constraints of the eyeball, and equivariance to viewpoint changes—all while generalizing across subjects, poses, and environments.

Despite recent progress, existing methods still face two key limitations: (1) **Lack of a unified physically grounded representation.** As illustrated in Fig. 1, appearance-based methods (Zhang et al., 2015; Chen & Shi, 2018; Cheng et al., 2020b) directly regress gaze from image features without explicit geometric constraints [Fig. 1(a)], leading to weak interpretability and sensitivity to appearance bias. Model-based methods (Chen et al., 2008; Hennessey et al., 2006; Świrski & Dodgson, 2013) reconstruct eyeball geometry [Fig. 1(b)] but rely on strong assumptions, personal

---

[1]Tianjin University, China [2]Tianjin Artificial Intelligence Innovation Center, China [3]Academy of Military Sciences (AMS), China [4]Zhengzhou University, China. Correspondence to: Erwei Yin <yinerwei1985@gmail.com>.

*Proceedings of the $43^{rd}$ International Conference on Machine Learning*, Seoul, South Korea. PMLR 306, 2026. Copyright 2026 by the author(s).

calibration, or specialized hardware. These two isolated methods fail to combine geometric constraints with structural modeling to form a unified representation, limiting generalization across subjects and devices. (2) **Limited generalization across viewpoints and domains.** Gaze estimation must operate under diverse head poses, camera viewpoints, and environments, yet existing models remain sensitive to such variations. Even with costly multi-view training, they struggle to bridge controlled laboratory and in-the-wild domains [Fig. 1(f)] due to the perspectives distribution shifts. This reflects the absence of rotation-equivariant and geometric–structural consistency—two critical inductive biases for generalizable gaze modeling.

To address these issues, we propose **SG-Gaze**, a framework that instantiates *physics-as-representation* by learning a **S**tructurally and **G**eometrically consistent **R**epresentation (**SGR**). SG-Gaze is built on a dual-branch architecture that jointly enforces geometric, structural, and equivariant priors—moving beyond the appearance-vs.-model dichotomy toward a unified representation grounded in gaze physics. Given a sequence of near-eye images, a shared backbone extracts visual features $F$, which are processed by two co-operative branches. The **A**nalytical **G**aze **E**stimation (AGE) branch encodes features onto a geodesically aligned spherical manifold [Fig. 1(c1)], explicitly enforcing the geometric constraint that gaze directions lie on a unit sphere. The **M**odel-**G**uided **R**econstruction (MGR) branch reconstructs parametric 3D eyeball structure under weak 2D edge supervision [Fig. 1(c2)], enforcing anatomical structure without dense 3D annotations. An adversarial alignment module bridges the two branches [Fig. 1(d)], yielding a unified SGR consistent with both appearance cues and physical priors. The notations $\text{Max}(F \cap G)$ and $\text{Max}(F \cap S)$ in the figures are conceptual annotations indicating that adversarial alignment extracts the maximally geometric-consistent $(G)$ and maximally structural-consistent subsets $(S)$ from $F$, rather than additional optimization operators. To further improve generalization under viewpoint changes, we introduce **V**iew-**C**onsistent **R**egularization (**VCR**), which augments SGR learning with synthetic viewpoint perturbations and enforces rotation-equivariant consistency across gaze vectors and structural projections [Fig. 1(e)]. This simulates multi-view conditions without costly data collection and mitigates controlled-to-real domain gaps by covering a larger field-of-view [Fig. 1(g)]. SG-Gaze jointly predicts 3D eyeball, gaze direction, and auxiliary 2D semantic projections within a coherent, physically grounded framework.

Comprehensive experiments on synthetic dataset UnityEyes (Wood et al., 2016) and real-world datasets TEyeD (Fuhl et al., 2021), LPW (Tonsen et al., 2016) show that SG-Gaze achieves state-of-the-art accuracy and strong cross-dataset generalization. We further quantitatively validate its physical grounding through high spherical correlation ($r =$

0.89), anatomical accuracy (e.g., $11.8 \pm 0.9$ mm eyeball radius), and low reprojection error(1.88 px). In summary, our main contributions are three-fold:

- We propose SG-Gaze, a unified dual-branch framework that integrates analytical gaze estimation and model-guided reconstruction based on physical principles of gaze. This design enables learning a Structurally and Geometrically Consistent Representation (SGR) that is interpretable and physically grounded.
- We introduce View-Consistent Regularization (VCR), a principled training strategy that enforces rotation-equivariant consistency between gaze vectors and structure projections under synthetic viewpoint perturbations, effectively alleviating domain distribution gaps.
- Experimental results show that SG-Gaze achieves state-of-the-art performance, with up to 38.61% baseline improvement across 12 challenging transfer scenarios without accessing target-domain data, backed by quantitative evidence of its physical plausibility.

## 2. Related Work

**Model-based Methods** Model-based approaches reconstruct eyeball's anatomical structure (Chen et al., 2008; Hennessey et al., 2006; Wood & Bulling, 2014) using infrared glints, corneal reflections (Hansen & Ji, 2009), or parametric 3D models (Liu et al., 2020; Popovic et al., 2023). Differentiable rendering (Zhao et al., 2020; Wang et al., 2024) and neural implicit surfaces (Ruzzi et al., 2023) have improved geometric fidelity and robustness. Recent works extend model-based estimation with representation learning and adaptability: $\text{De}^2\text{Gaze}$ (Xiao et al., 2025) proposed a 3D eye tracking method using deformable and decoupled representations with lightweight time-series. Few-shot personalization (Wu et al., 2024) and uncertainty-aware fitting (Zhong et al., 2024) further enhance cross-subject adaptability.

**Appearance-based Methods** Deep learning has driven most recent progress in gaze estimation, leveraging large-scale datasets (Krafka et al., 2016; Zhang et al., 2020; 2015; Kellnhofer et al., 2019; Zhang et al., 2017). Models predictions from eye crops (Zhang et al., 2015; Cheng et al., 2020b), full-face images (Bao et al., 2021; Balim et al., 2023; Chen & Shi, 2018; Cheng et al., 2020a; Krafka et al., 2016; Zhang et al., 2017), or their fusion (Krafka et al., 2016; Park et al., 2020). Leveraging two-eye asymmetry has been shown to improve gaze prediction accuracy (Chen & Shi, 2018; Cheng et al., 2020b). Recent approaches employ attention mechanisms, hierarchical and multi-stream architectures, as well as transformer-based backbones (Qin et al., 2025; Yin et al., 2024).

**Cross-domain Gaze Estimation** Existing methods include unsupervised domain adaptation (Lahiri et al., 2018; Zhang et al., 2022a), contrastive objectives (Wang et al., 2022)

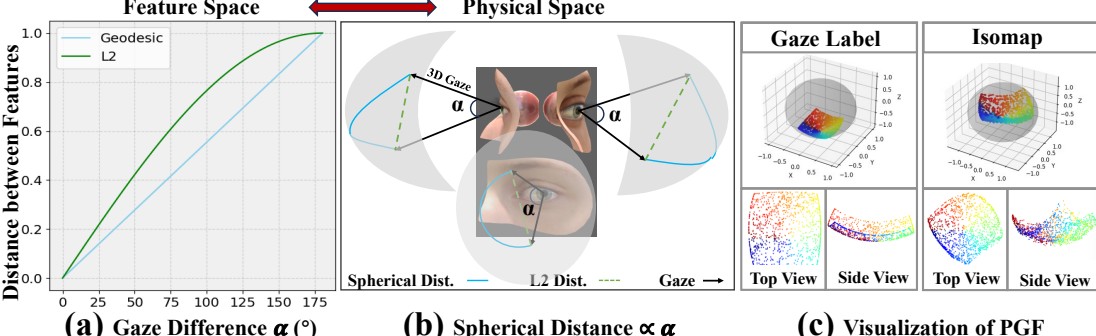

*Figure 2.* Physical geometry analysis of gaze features. (a) L2 and geodesic distances vs. angular differences on UnityEyes. (b) Geodesic distance on feature sphere proportional to gaze differences. (c) Visualization of the principal gaze feature after Isomap projection, where features exhibit a spherical distribution consistent with gaze labels, validating the geometric consistency of our representation.

and collaborative learning with outlier guidance (Liu et al., 2021). But these methods require target-domain data. Techniques include rotation-consistent regularization (Bao et al., 2022), source-domain feature purification (Cheng et al., 2022), style perturbation (Zhou et al., 2024), and synthetic imagery (Zhang et al., 2022b). Recent work incorporates geometric constraints and 3D structure. 3D eye mesh regression (Ververas et al., 2024) and geometry projections with spherical training (Bao & Lu, 2024) enhance cross-domain robustness without target-domain access.

## 3. Structural and Geometric Consistency

**Gaze Fact** Our gaze representation is grounded in two fundamental physical properties of human vision.

**(1) Gaze vectors lie on a unit 3D sphere.** Gaze direction can be defined as a unit vector from the eyeball center to a fixation point. Hence, all gaze vectors reside on the surface of the unit sphere $\mathbb{S}^2$. As illustrated in Fig. 2(b), the geodesic distance on $\mathbb{S}^2$ directly corresponds to the angular difference between gaze directions, defining a spherical manifold structure for gaze representation.

**(2) Eyeball exhibits strong anatomical constraints.** The eyeball approximates a rigid 3D sphere, where the gaze direction is determined by the geometric relationship between the eyeball center and the pupil center. This implies that accurate gaze estimation should not rely solely on image pixels or implicit features, but must respect anatomical plausibility and strong structural constraints.

**Spherical Structure in Feature Space** To examine whether feature space preserves geometric structure of gaze, we analyze on UnityEyes (Wood et al., 2016). We pretrain a ResNet-18 (He et al., 2016) feature extractor $F_{\theta_1}(\cdot)$ with $L_1$ loss: $f_i = F_{\theta_1}(x_i)$, $\min_{\theta_1, \theta_2} \sum_{i=1}^{N} L_1(y_i, L_{\theta_2}(f_i))$, where $f_i \in \mathbb{R}^{512}$, $x_i$ is eye image and $y_i$ is 3D Cartesian gaze vector, $L_{\theta_2}(\cdot)$ is the final fully connected layer. After training converges, both $F_{\theta_1}$ and $L_{\theta_2}$ are frozen. We measure geodesic distance (a $k$-nearest neighbor graph over

the 512D embeddings using Euclidean distance) and compare them with the angular differences of their ground-truth gaze vectors. **Key Observation**: The two distances exhibit a strong linear correlation (Pearson's $r = 0.89$), indicating that the feature space implicitly preserves the spherical topology of gaze directions. Motivated by this, we follow AGG (Bao & Lu, 2024) and extract *Principal Gaze Feature (PGF)* by applying Isometric Mapping (Isomap) (Tenenbaum et al., 2000) to project $f_i$ into a 3D subspace. As shown in Fig. 2(c). The PGFs lie on the surface of a 3D sphere, preserving the structural geometry of gaze.

These complementary physical principles motivate reformulating gaze estimation as a representation learning problem under explicit structural and geometric constraints. We advocate learning a unified Structurally and Geometrically Consistent Representation **(SGR)**. Formally, we define the SGR $\in \mathbb{R}^d$ to satisfy three key physical constraints:

**(1) Geometric Consistency** SGR's 3D projection must lie on the unit sphere ($\|\text{SGR}_{\text{proj}}\| = 1$), preserving geodesic locality in $\mathbb{S}^2$ and the angular structure of gaze space: $d_g(\mathbf{z}_i, \mathbf{z}_j) \propto \angle(\mathbf{g}_i, \mathbf{g}_j)$.

**(2) Structural Consistency** SGR must enable 3D eyeball reconstruction through a differentiable mapping: $\|\text{Proj}(D_e(\mathbf{z})) - \text{Edge}(I)\|_2$, where $D_e(\cdot)$ decodes eyeball parameters and $\text{Proj}(\cdot)$ is 3D-to-2D projection, ensuring it corresponds to physically possible eyeball.

**(3) Rotation Equivariance** SGR must be consistent with 3D viewpoint changes: $D_g(T_P(\text{SGR})) \approx P \cdot D_g(\text{SGR})$, where $D_g(\cdot)$ is the gaze decoder. $T_P$ is a learned rotation operator that approximates the physical 3D rotation P.

## 4. Methodology

### 4.1. Overview

The overall pipeline of SG-Gaze is illustrated in Fig. 3. Given a sequence of near-eye image frames $I = \{I_1, \ldots, I_n\}$, the framework $\Phi$ consists of three core components: **(1) Base Branch:** Input frames $I$ are encoded

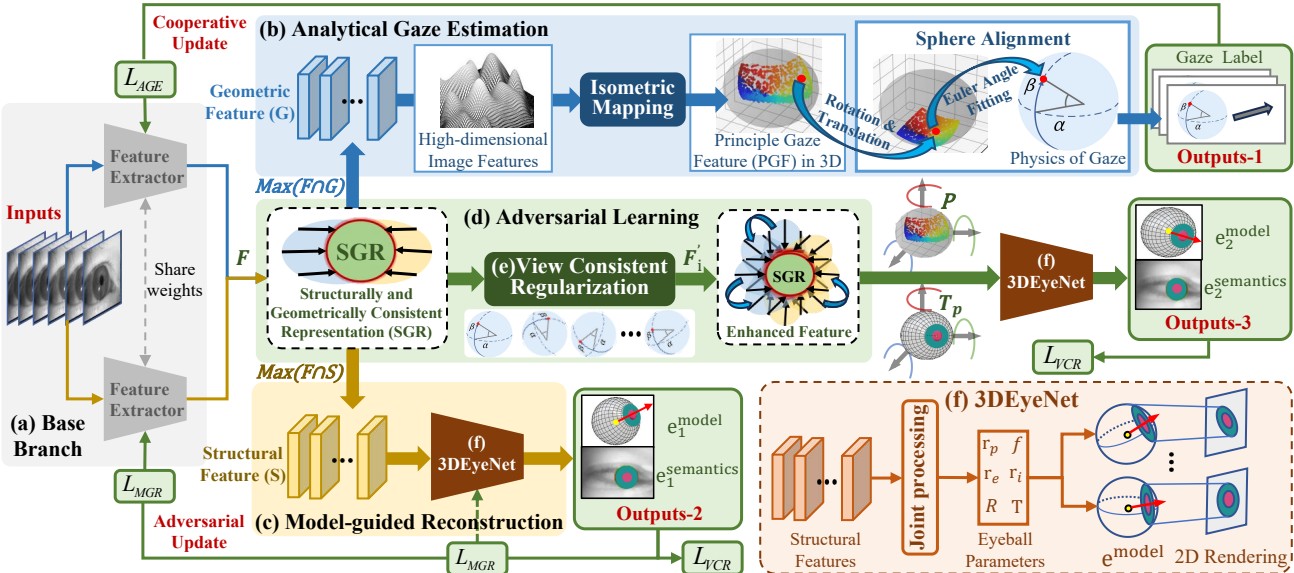

*Figure 3.* Overview of SG-Gaze. The pipeline consists of a shared feature extractor, an Analytical Gaze Estimation (AGE) branch, a Model-Guided Reconstruction (MGR) branch, adversarial alignment between branches, and a View-Consistent Regularization (VCR) module. 3DEyeNet is used to implement differentiable rendering for anatomically plausible eyeball modeling.

by a shared feature extractor $B$ (ResNet-18) into visual features $F = B(I)$ [Fig. 3(a)]. **(2) Dual-Branch Decoding with Adversarial Learning:** The decoder $D$ contains two cooperative branches: the Analytical Gaze Estimation **(AGE)** branch [Fig. 3(b)] projects features onto a geodesically aligned spherical manifold to yield an interpretable embedding for gaze regression; the Model-Guided Reconstruction **(MGR)** branch [Fig. 3(c)] predicts 3D eyeball parameters (iris, pupil, sphere center) and jointly optimizes the gaze via physical modeling constraints through 3DEyeNet. Both branches share intermediate representation and are optimized jointly. A cross-task adversarial discriminator in $\Phi_{\text{afr}}$ [Fig. 3(d)] encourages AGE and MGR to produce structurally and geometrically consistent representation (SGR). **(3) View-Consistent Regularization (VCR):** The VCR module [Fig. 3(e)] applies synthetic rotations and enforces rotation-equivariant consistency of SGR across gaze vectors and structural projections, enhancing robustness to viewpoint variation without multi-view data. Through joint optimization, SG-Gaze learns a unified SGR that seamlessly integrates appearance cues with physical modeling. It outputs the optimized 3D eyeball model $e^m$, gaze vector $e^g$, and 2D semantic projections $e^s$ with improved accuracy and interpretability. Formalized as $\{e^m, e^g, e^s\} = \Phi(I)$.

## 4.2. Analytical Gaze Estimation

The AGE branch explicitly encodes the geometric structure of gaze directions into representation learning process. Unlike black-box regression, AGE leverages spherical projection and analytical fitting to construct gaze vectors that inherently respect the 3D spherical topology.

**Spherical Projection** Motivated by the observed correlation between feature geodesic distance and gaze angular difference (Section 3), we apply Isomap to project features $\mathbf{f}_i \in \mathbb{R}^{512}$ into a 3D Principal Gaze Feature (PGF) space: $\mathbf{p}_i\}_{i=1}^{N'} = \text{Isomap}(\{f_i\}_{i=1}^{N'})$, with detailed implementation provided in Section 5.4 (2). The resulting PGFs reside on a 3D sphere, preserving the gaze-aware topology [Fig. 2(c)].

**Spherical Fitting** To obtain gaze directions analytically, we adopt a Spherical Fitting (SF) procedure following AGG (Bao & Lu, 2024). Specifically, we estimate the sphere center $O_c$ and apply a rotation $R$ to align the coordinate system: $\mathbf{p}_i' = R(\mathbf{p}_i - O_c) = (x_i', y_i', z_i')^\top$. Euler angles $(\theta_i', \psi_i')$ are then derived from the rotated point $p_i'$ via physically interpretable mappings: $\theta_i' = k_1 \arctan\left(\frac{x_i'}{z_i'}\right) + b_1$, $\psi_i' = k_2 \arcsin(y_i') + b_2$, where $(k_1, k_2, b_1, b_2)$ are learnable scale and bias parameters. The final 3D gaze vector is: $y_i' = \text{SF}_{\theta_s}(p_i)$, $\theta_s = \{O_c, R, k_1, k_2, b_1, b_2\}$. Parameters $\theta_s$ are optimized by minimizing angular error: $\min_{\theta_s} \sum_{i=1}^N \mathcal{L}_{\text{angular}}(\mathbf{y}_i, \text{SF}_{\theta_s}(\mathbf{p}_i))$.

The AGE branch ensures that local variations in representation correspond to proportional changes in gaze direction. Unlike AGG which uses Isomap for final regression layer replacement, our AGE creates an intermediate geometry-consistent representation that serves as a geometric prior and facilitates adversarial alignment with the MGR branch.

## 4.3. Model-Guided Reconstruction

While the AGE enforces geometric alignment, it ignores anatomical feasibility constraints. The MGR branch addresses this by imposing a structured, physically grounded

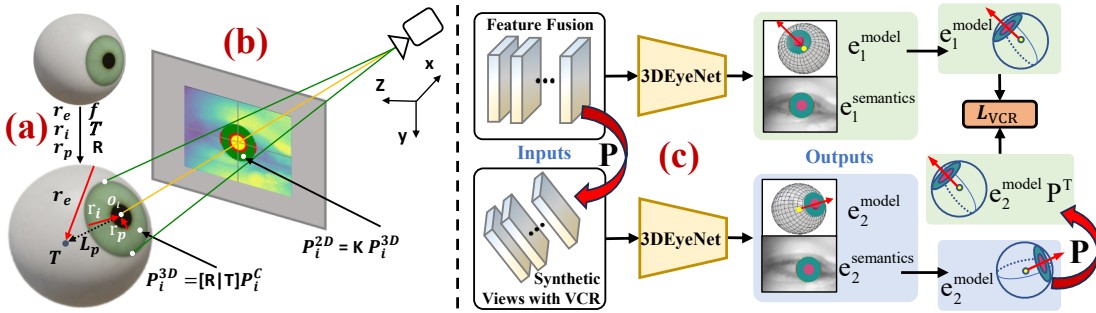

*Figure 4.* Structure-aware modeling and view-consistent regularization. (a) The parametric eyeball is reconstructed in camera coordinate system using the predicted parameters, demonstrating structural consistency. (b) Projection of the iris and pupil regions onto 2D plane, yielding the corresponding semantic masks through weak supervision of sparse edge points. (c) VCR: Enforcing consistency in gaze prediction space and structural edge projections under simulated 3D rotations.

prior through parametric 3D eyeball reconstruction.

**Parametric Eyeball Modeling** As shown in Fig. 4(a). Following established anatomical eye models (Świrski & Dodgson, 2013; Dierkes et al., 2018; Popovic et al., 2023; Xiao et al., 2025), for each input frame $I$, the eyeball is represented as a sphere with parameters $\mathbf{E}_{\text{para}} = \{\mathbf{r_e}, \mathbf{r_i}, \mathbf{T}, \mathbf{r_p}, \mathbf{R}\}$, where $r_e, r_i, r_p$ denote eyeball, iris and pupil radii. $T \in \mathbb{R}^3$ indicates the eyeball center in the camera coordinate system, and $R \in SO(3)$ describes its orientation. Under a pinhole camera assumption with shared focal length $f_x = f_y = f$, the optical axis is estimated as $\frac{o_i - o_e}{\|o_i - o_e\|}$, with $o_e$ and $o_i$ denoting the eyeball and iris center. While anatomical deviations exist (e.g., corneal protrusion), the dominant spherical geometry, combined with limited pupil movement ranges, provides a strong structural prior.

**Deformation and Projection** Following De²Gaze (Xiao et al., 2025), pupil and iris point clouds are generated in the canonical space as concentric disks and rings: $P_p^C = \{(r_p \rho \cos(\theta), r_p \rho \sin(\theta), -L_p) \mid \rho \in [0,1], \theta \in [0,2\pi]\}, P_i^C = \{(r \cos(\theta), r \sin(\theta), -L_p) \mid r = r_p + \rho(r_i - r_p), \rho \in [0,1], \theta \in [0,2\pi]\}$. To render the semantic supervision signals, we transform the canonical point clouds into the camera coordinate system by applying the predicted rotation $R$ and translation $T$, and project them onto 2D image with intrinsic $K$: $P_p^{2D} = K[R|T]P_p^C$, $P_i^{2D} = K[R|T]P_i^C$. As shown in Fig. 4(b), only iris/pupil edges are supervised to reduce cost. Unlike De²Gaze which focuses on decoupled representation for temporal tracking, our MGR branch serves as a structural inductive bias that enforces physically grounded predictions, interpretable feedback, and robustness to sparse supervision and domain shifts. While the AGE branch can implicitly compensate for systematic biases not captured by the MGR's physical model.

### 4.4. View-Consistent Regularization

Unlike 3DGazeNet (Ververas et al., 2024) which enforces multi-view consistency using explicit multi-view data, as shown in Fig. 4(c), we introduce View-Consistent Regularization (VCR) to impose rotation-equivariant constraints without multi-view input. VCR treats viewpoint changes as known elements in $SO(3)$ and enforces their consistency across feature, prediction, and structural spaces, serving as a principled inductive bias for unseen viewpoints.

**(1) Feature Rotation** Given an encoder feature $\mathbf{f}_i \in \mathbb{R}^d$, a known gaze-space rotation $P \in SO(3)$ is mapped to the feature-space via a learned linear operator $W \in \mathbb{R}^{d \times 3}$: $T_p = WPW^\dagger, f_i' = T_p f_i$, $W^\dagger$ denotes the pseudo-inverse, ensuring that the feature-space rotation $T_p$ approximates the geometric transformation of the gaze-space rotation $P$.
**(2) Geometric Prediction Consistency** Given original features $f_i$ and rotated features $T_p(f_i)$, the gaze decoder $D_g$ produces: $g_i = D_g(f_i)$ and $\mathbf{g}_i' = D_g(T_p(f_i))$. We enforce consistency through: $\mathbf{g}_i' \approx P\mathbf{g}_i$, ensuring that feature-level rotations commute with physical rotations in gaze space.
**(3) Structural Projection Consistency** The eyeball parameters decoder $D_e$ produces: $e_i = D_e(f_i)$ and $e_i' = D_e(T_p(f_i))$. Through MGR, the canonical 3D pupil/iris point clouds $P_p^{3D}/P_i^{3D}$ generated by $e_i$ both undergo the same rotational $P$, and $e_i$ generates 3D pupil/iris point clouds $\widetilde{P}_p^{3D}/\widetilde{P}_i^{3D}$. All 3D point clouds are projected with camera intrinsics $K$ for 2D supervision, and constrained to remain consistent under rotation: $(P_p^{2D} = KPP_p^{3D}) \approx (\widetilde{P}_p^{2D} = K\widetilde{P}_p^{3D})$, $(P_i^{2D} = KPP_i^{3D}) \approx (\widetilde{P}_i^{2D} = K\widetilde{P}_i^{3D})$. By jointly constraining the above consistency, VCR enforces comprehensive rotation-equivariance across the entire representation pipeline, significantly enhancing cross-view generalization while maintaining physical plausibility.

### 4.5. Loss

**(1) AGE Branch Loss** The AGE branch is supervised by angular error between predicted and ground-truth gaze directions. For $N$ samples with weight $w_{\text{AGE}}$, we defined:

$$L_{AGE} = w_{\text{AGE}} \frac{1}{N} \sum_{n=1}^{N} \arccos \left( \frac{\hat{\mathbf{g}}_{AGE}^n \cdot \mathbf{g}_{GT}^n}{\|\hat{\mathbf{g}}_{AGE}^n\|\|\mathbf{g}_{GT}^n\|} \right). \quad (1)$$

*Table 1.* Comparison of SG-Gaze with baseline models under cross-dataset ($D_{T_1}, D_{T_2}, D_S, D_L$) and within-dataset ($D_{T_1}, D_{T_2}$) evaluation settings. 3D Gaze error is in degrees (lower is better, ▼ denotes reduction, ▲ denotes increase).

| Method | Source: $D_{T_1}$ | | | Source: $D_{T_2}$ | | | Within-dataset | |
|---|---|---|---|---|---|---|---|---|
| | $\to D_S$ | $\to D_L$ | $\to D_{T_2}$ | $\to D_S$ | $\to D_L$ | $\to D_{T_1}$ | $D_{T_1}^*$ | $D_{T_2}^*$ |
| ResNet-18 | 5.02 | 6.83 | 3.20 | 5.31 | 6.68 | 3.29 | 1.68 | 1.59 |
| ResNet-18+SG-Gaze | 4.10▼18.32% | 5.07▼25.77% | 2.07▼35.31% | 3.93▼25.99% | 4.87▼27.10% | 2.03▼38.30% | 1.11▼33.93% | 1.33▼16.35% |
| ResNet-50 | 4.04 | 5.47 | 2.54 | 5.16 | 5.14 | 3.16 | 1.32 | 1.34 |
| ResNet-50+SG-Gaze | 3.91▼3.21% | 3.75▼31.44% | 1.88▼25.78% | 4.36▼15.50% | 4.21▼18.09% | 1.94▼38.61% | 1.22▼7.58% | 1.09▼18.66% |
| VGG-16 | 5.5 | 7.14 | 3.61 | 5.94 | 7.61 | 3.94 | 2.12 | 2.35 |
| VGG-16+SG-Gaze | 5.13▼6.73% | 5.2▼27.02% | 3.30▼8.59% | 5.27▼11.28% | 5.33▼29.97% | 2.97▼24.62% | 2.78▲31.13% | 3.13▲33.19% |
| Swin-T | 4.01 | 4.92 | 2.41 | 4.88 | 4.85 | 2.92 | 1.19 | 1.18 |
| Swin-T+SG-Gaze | 3.82▼4.74% | 3.71▼24.59% | 1.85▼23.24% | 4.31▼11.68% | 4.11▼15.26% | 1.91▼34.59% | 1.07▼10.08% | 1.01▼14.41% |
| DINOv2 (ViT-S/14) | 3.96 | 4.58 | 2.28 | 4.71 | 4.62 | 2.58 | 1.24 | 1.22 |
| DINOv2 (ViT-S/14)+SG-Gaze | 3.85▼2.78% | 3.67▼19.87% | 1.82▼20.18% | 4.28▼9.13% | 4.14▼10.39% | 1.88▼27.13% | 1.17▼5.65% | 1.04▼14.75% |
| DINOv2 (ViT-B/14) | 3.89 | 4.21 | 2.09 | 4.53 | 4.39 | 2.35 | 1.04 | 1.07 |
| DINOv2 (ViT-B/14)+SG-Gaze | 3.74▼3.86% | 3.52▼16.39% | 1.76▼15.79% | 4.15▼8.39% | 4.01▼8.66% | 1.82▼22.55% | 0.96▼7.69% | 0.94▼12.15% |

**(2) MGR Branch Loss** The MGR branch is supervised by both structural projection and gaze consistency. For $N$ frames and $K$ edge points with weights $w_1$ and $w_2$:

$$L_{MGR}^{edge} = w_1 \frac{1}{NK} \sum_{n=1}^{N} \sum_{k=1}^{K} \|P_{nk}^{2D} - P_{n\,\mathrm{argmin}_j \|P_{nk}^{2D} - P_{nj}^{GT,2D}\|}^{GT,2D}\|,$$

$$L_{MGR}^{gaze} = w_2 \frac{1}{N} \sum_{n=1}^{N} \arccos\left(\frac{\hat{\mathbf{g}}^n \cdot \mathbf{g}^n}{\|\hat{\mathbf{g}}^n\|\|\mathbf{g}^n\|}\right).$$
$$(2)$$

**(3) Adversarial Loss** The discriminator $D$ is a 3-layer MLP with a gradient reversal layer, trained to distinguish $f_A$ and $f_M$ from two branches, while the AGE and MGR branches are optimized to maximize the discriminator loss, encouraging branch-invariant representations.

$$L_{adv} = E_{f_A}[\log D(f_A)] + E_{f_M}[\log(1 - D(f_M))], \quad (3)$$

where $f_A = h_A(F) \in \mathbb{R}^{512}$ denotes the geometry-aware feature, and $f_M = h_M(F) \in \mathbb{R}^{512}$ denotes the structure-aware feature, where $F$ is the shared backbone feature.

**(4) VCR Loss** Following Sec 4.4, VCR enforces rotation-equivariant consistency under a known rotation $P \in \mathbb{R}^{3 \times 3}$:

$$L_{VCR}^{gaze} = w_{gaze}' \frac{1}{N} \sum_{i=1}^{N} \|\mathbf{g}_i' - P\mathbf{g}_i\|_2^2,$$

$$L_{VCR}^{edge} = w_{proj}' \frac{1}{N} \sum_{i=1}^{N} \left(\|P_p^{2D} - \widetilde{P}_p^{2D}\|_2^2 + \|P_i^{2D} - \widetilde{P}_i^{2D}\|_2^2\right).$$
$$(4)$$

# 5. Experiments

## 5.1. Datasets and Preprocessing

**Dataset** We evaluate on two real-world datasets and one synthetic dataset: **(1) TEyeD**(Fuhl et al., 2021) contains 20M+ images from 132 subjects with 2D/3D landmarks, segmentation, gaze vectors and eye movement labels. We adopt three subject-disjoint splits: $D_{T_1}$ (200k/30k, 16 subjects), $D_{T_2}$ (200k/30k, 16 subjects), and $D_S$ (50k/12k, 39 subjects), supporting both within- and cross-subset evaluation. **(2) LPW** (Tonsen et al., 2016) is collected in daily-life conditions with large illumination, pose, and occlusion variations. $D_L$: We sample 97k/27k images from 22 subjects, excluding blurred or occluded frames, and use it as a target domain for cross-dataset evaluation under weak 2D edge supervision. **(3) UnityEyes** (Wood et al., 2016) is a synthetic dataset with accurate 3D gaze and structure annotations. We use (160k/40k) images from 20 virtual subjects for backbone pretraining, spherical feature analysis, and rotation-consistency regularization.

**Implementation Details** Eye images are cropped to $320 \times 240$ and normalized. Data augmentation includes Gaussian blur (std 1.0–2.0), random noise (0–30%), and horizontal flipping (20%). In the MGR branch, weak structural supervision is provided via sparse iris and pupil edges: 26 iris edge points are uniformly sampled over $\theta \in [0, 0.1\pi] \cup [0.9\pi, 1.1\pi] \cup [1.9\pi, 2\pi]$ with radius $\rho \in [0, 1]$, while 128 pupil contour points are sampled over $[0, 2\pi]$. Isomap is implemented using Scikit-learn with 300 nearest neighbors. Optimization uses LAMB (You et al., 2019) with initial learning rate $2 \times 10^{-3}$ and weight decay 0.02.

## 5.2. Main Results

**(1) Cross-dataset Evaluation** We evaluate SG-Gaze across domains and backbones (Table 1). With ResNet-18, SG-Gaze cuts gaze error by up to **35.31%** in $D_{T_1} \to D_{T_2}$ and over **25%** on most tasks; ResNet-50 + SG-Gaze yields **26.77%–38.61%** gains. We further validate scalability on modern Transformer encoders (DINOv2 ViT-S/14/B/14, Swin-T), with consistent improvements across transfers.

*Table 2.* Cross-domain comparison with state-of-the-art gaze estimation methods. 3D Gaze error is in degrees (lower is better). SG-Gaze achieves the best generalization across unseen domains, benefiting from its structurally and geometrically consistent representation.

| Method | $D_{T_1} \to D_S$ | $D_{T_1} \to D_{T_2}$ | $D_{T_1} \to D_L$ | $D_{T_2} \to D_{T_1}$ | $D_{T_2} \to D_S$ | $D_{T_2} \to D_L$ | $D_S \to D_{T_1}$ | $D_S \to D_{T_2}$ | $D_S \to D_L$ | $D_L \to D_{T_1}$ | $D_L \to D_{T_2}$ | $D_L \to D_S$ |
|---|---|---|---|---|---|---|---|---|---|---|---|---|
| RAT (Bao et al., 2022) | 5.82 | 2.83 | 5.73 | 3.91 | 5.31 | 5.97 | 6.18 | 5.29 | 4.79 | 5.68 | 4.59 | 4.83 |
| Latentgaze (Lee et al., 2022) | 5.5 | 3.14 | 5.81 | 3.61 | 5.94 | 5.83 | 5.11 | 4.99 | 4.83 | 5.12 | 4.35 | 5.02 |
| FFGaze (Zhang et al., 2017) | 5.04 | 3.47 | 5.44 | 3.54 | 5.16 | 5.44 | 4.96 | 5.16 | 5.24 | 5.32 | 4.34 | 4.77 |
| PureGaze (Cheng et al., 2022) | 4.5 | 2.94 | 4.90 | 2.61 | 4.94 | 5.71 | 4.61 | 5.94 | 5.21 | 5.03 | 4.35 | 4.97 |
| AGG (Bao & Lu, 2024) | 4.0 | 2.21 | 4.33 | 2.15 | 4.21 | 4.94 | 3.44 | 4.53 | 4.21 | 4.02 | 3.58 | 4.44 |
| De²Gaze (Xiao et al., 2025) | 4.9 | 2.88 | 5.10 | 3.22 | 5.02 | 5.88 | 4.79 | 5.25 | 4.97 | 5.10 | 4.72 | 5.33 |
| Baseline | 5.02 | 3.20 | 6.83 | 3.29 | 5.31 | 6.68 | 5.21 | 5.92 | 5.41 | 5.32 | 4.77 | 5.63 |
| ResNet18+SG-Gaze | 4.10 | 2.07 | 5.07 | 2.03 | **3.93** | 4.87 | 3.41 | 4.36 | 4.43 | 4.12 | **3.24** | 4.23 |
| ResNet50+SG-Gaze | **3.91** | **1.88** | **3.75** | **1.94** | 4.36 | **4.21** | **3.29** | **4.04** | **4.18** | **3.87** | 3.35 | **4.01** |

*Table 3.* The table reports backbone, Params, FLOPs and loss settings. SG-Gaze achieves the best trade-off between accuracy, efficiency and semantic consistency without domain shift.

| Methods | Backbone | Params /(M) | FLOPs /(G) | Loss | 3D gaze [°]↓ | 2D gaze [°]↓ | Sem. Iou | 2D eye cent.[px]↓ |
|---|---|---|---|---|---|---|---|---|
| (Fuhl et al., 2021) | ResNet50 | 26.08 | 24.97 | Gaze | 1.88 | 6.90 | N/A | N/A |
| (Kim et al., 2019) | CNN | 0.16 | 0.14 | Gaze | 3.65 | 8.34 | N/A | N/A |
| Transformer-based | ResNet18 | 15.53 | 14.83 | Gaze | 1.57 | 6.36 | N/A | N/A |
| (Vaswani, 2017) | ResNet50 | 27.66 | 24.98 | Gaze | 1.77 | 6.40 | N/A | N/A |
| QueryDETR | ResNet18 | 16.36 | 18.33 | Gaze | 3.12 | 8.01 | N/A | N/A |
| (Carion et al., 2020) | ResNet50 | 29.89 | 28.42 | Gaze | 3.08 | 7.86 | N/A | N/A |
| Nikola et al. | ResNet50 | 28.56 | 26.44 | Gaze | 1.04 | 7.40 | N/A | N/A |
| | ResNet50 | 28.56 | 26.44 | Sem. | 20.16 | 39.10 | 92.5% | 11.41 |
| (Popovic et al., 2023) | ResNet50 | 28.56 | 26.44 | S+G+C | 1.21 | 10.39 | 91.4% | 2.02 |
| De²Gaze | ResNet18 | 14.48 | 13.56 | Gaze | **0.54** | 5.43 | N/A | N/A |
| | ResNet18 | 14.48 | 13.56 | Sem. | 20.12 | 39.09 | **94.2%** | 11.41 |
| (Xiao et al., 2025) | ResNet18 | 14.48 | 13.56 | S+G+C | 0.96 | 7.6 | 93.4% | **1.52** |
| SG-Gaze (Ours) | ResNet18 | **11.68** | **11.24** | Gaze | 0.94 | 6.22 | N/A | N/A |
| | ResNet18 | 11.68 | 11.24 | Sem. | 21.12 | 39.69 | 93.3% | 12.22 |
| | ResNet18 | 11.68 | 11.24 | S+G+C | 1.11 | 8.82 | 92.1% | 1.88 |

This confirms physical priors complement strong pre-trained features: backbones provide rich semantics, while SG-Gaze adds geometric-structural consistency absent in pre-training. SG-Gaze intentionally enforces stronger physical inductive biases for structural and cross-view consistency, slightly hurting fine-grained fitting in low-capacity models (e.g., VGG-16) but substantially boosting cross-domain performance. This trade-off reflects real-world requirements where domain shift is inevitable, demonstrating gaze generalization, independent of backbone choice or domain shift.

**(2) Comparison with SOTA Methods** We benchmark SG-Gaze against recent generalizable approaches, including RAT (Bao et al., 2022), LatentGaze (Lee et al., 2022), FFGaze (Zhang et al., 2017), De²Gaze (Xiao et al., 2025), AGG (Bao & Lu, 2024) and PureGaze (Cheng et al., 2022). As shown in Table 2, SG-Gaze achieves the best accuracy in nearly all 12 cross-domain transfer tasks. ResNet-50 + SG-Gaze reduces error to **1.88°** on $D_{T_1} \to D_{T_2}$ and **1.94°** on $D_{T_2} \to D_{T_1}$, outperforming the strongest baseline (AGG) by an average of **7.7%**. SG-Gaze also shows clear advantages in challenging scene transfers such as $D_S \to D_{T_1}$ and $D_S \to D_{T_2}$, stressing that SG-Gaze not only enhances backbone models but also surpasses existing SOTA methods.

**(3) Within-dataset Evaluation** We further evaluate SG-

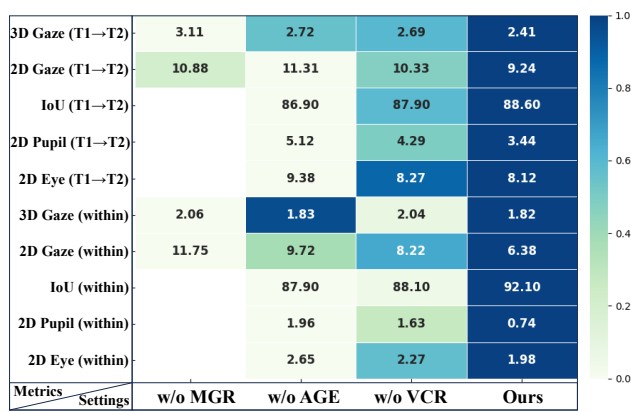

*Figure 5.* Ablation study heatmap showing the impact of removing different modules (AGE, MGR, VCR) on gaze and semantic metrics, Darker color indicates better performance.

Gaze under within-domain settings. As shown in Table 3, SG-Gaze achieves competitive performance (0.94° with ResNet18), while maintaining high efficiency (11.68M parameters, 11.24G FLOPs). Notably, it also improves semantic edge IoU and 2D eyeball center localization, indicating enhanced geometric and structural consistency. Our work prioritizes cross-domain generalization and efficiency over within-domain leaderboard performance, addressing the critical challenge of domain shift in real-world applications.

### 5.3. Ablation Studies

We conduct ablations on two TEyeD subsets ($D_{T_1}$, $D_{T_2}$) to assess the contribution of each component by selectively removing or replacing individual modules (Table 4).

**(1) Effect of Analytical Gaze Estimation** Removing the AGE branch ($w/o$ AGE) eliminates physically interpretable analytical supervision, increasing error to **2.57°** on $D_{T_1} \to D_{T_2}$. While semantic and center-based constraints still offer auxiliary guidance, the model loses explicit alignment between gaze predictions and analytical eye geometry, confirming the importance of geometric priors.

**(2) Effect of Model-Guided Reconstruction** Removing the MGR branch ($w/o$ MGR) leads to the largest performance drop, since 3D edge projection loss and the structural cues from iris/pupil are absent. In $D_{T_1} \to D_{T_2}$, the 3D gaze error increases from **1.88°** to **3.11°**. This confirms

*Table 4.* Ablation study. We compare the full SG-Gaze model with variants removing AGE, MGR, or VCR. The complete model achieves the best performance across all metrics, demonstrating that each component contributes positively to improved accuracy and robustness.

| Settings | Loss | $D_{T_1} \rightarrow D_{T_2}$ | | | | | within $D^*_{T_1}$ | | | | |
|---|---|---|---|---|---|---|---|---|---|---|---|
| | | 3D gaze [°]↓ | 2D gaze [°]↓ | Sem. Iou | 2D pupil cent.[px]↓ | 2D eye cent.[px]↓ | 3D gaze [°]↓ | 2D gaze [°]↓ | Sem. Iou | 2D pupil cent.[px]↓ | 2D eye cent.[px]↓ |
| w/o MGR Module | Gaze | 3.11 | 10.88 | N/A | N/A | N/A | 2.86 | 11.75 | N/A | N/A | N/A |
| w/o AGE Module | Gaze | 2.57 | 9.76 | N/A | N/A | N/A | 1.83 | 10.32 | N/A | N/A | N/A |
| | Sem. + Gaze + Cent. | 2.72 | 11.31 | 86.9% | 5.12 | 9.38 | 2.54 | 9.72 | 87.9% | 1.96 | 2.65 |
| w/o VCR | Gaze | 2.07 | 9.36 | N/A | N/A | N/A | 1.53 | 8.12 | N/A | N/A | N/A |
| | Sem. + Gaze + Cent. | 2.69 | 10.33 | 87.9% | 4.29 | 8.27 | 2.04 | 9.22 | 88.1% | 1.63 | 2.27 |
| SG-Gaze (**Ours**) | Gaze | **1.88** | **8.02** | N/A | N/A | N/A | **0.94** | **6.22** | N/A | N/A | N/A |
| | Sem. + Gaze + Cent. | 2.41 | 9.24 | **88.6%** | **3.44** | **8.12** | 1.11 | 8.82 | **92.1%** | **0.74** | **1.98** |

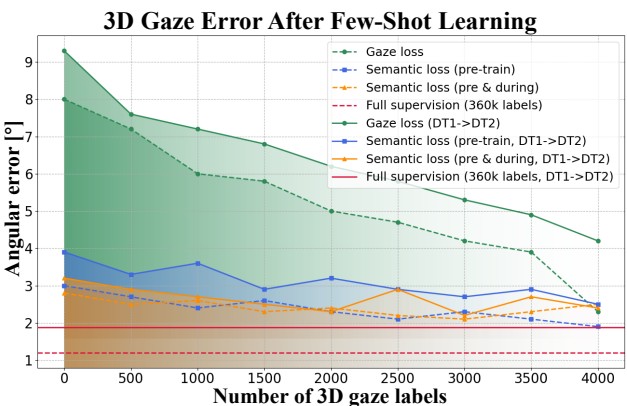

*Figure 6.* Few-shot fine-tuning results showing that semantic pre-training enables consistent performance gains within and across datasets, even with very limited gaze annotations.

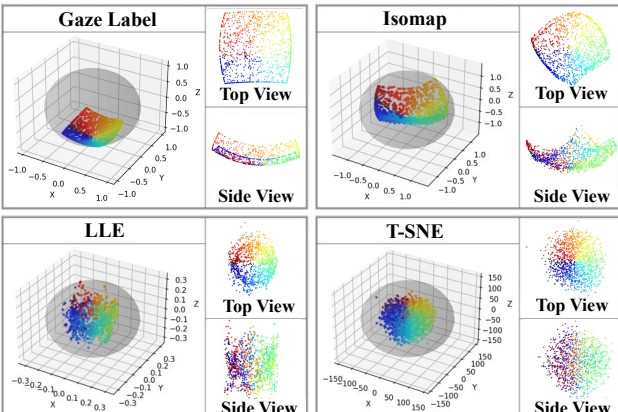

*Figure 7.* 3D visualization of gaze representations after dimensionality reduction (Isomap, t-SNE, and LLE). only Isomap preserves the global manifold and aligns well with GT gaze directions.

that enforcing edge-based structural constraints is crucial for geometric consistency and robustness.

**(3) Effect of View-Consistent Regularization** Discarding VCR (*w/o* VCR) weakens robustness to viewpoint changes, increasing cross-domain error from **1.88°** to **2.07°**. This shows that rotation-equivariant regularization not only augments the training distribution with view perturbations but also enhances cross-view generalization.

**(4) Ablation Heatmap** As shown in Fig. 5, the complete model consistently outperforms all ablated variants across metrics. Improvements in semantic IoU and 2D localization further indicate that joint multi-task training strengthens both geometric fidelity and physical consistency.

## 5.4. Efficient Learning under Limited Supervision

**(1) Few-Shot Adaptation** Accurate gaze estimation is limited by scarce 3D gaze annotations (Kim et al., 2019; Fuhl et al., 2021), while iris and pupil masks are abundant and easier to label. We adopt a two-stage strategy: pre-training on large-scale semantic labels followed by fine-tuning with few 3D gaze samples. As shown in Fig. 6, training from scratch with few annotations performs poorly, whereas se-

mantic pre-training yields a reasonable zero-shot baseline. Notably, fine-tuning with only 1% labeled gaze samples per subject achieves the largest performance gain.

**(2) Isomap Selection and Isometric Propagator** We evaluate three dimensionality reduction methods to project features $f_i$ into 3D space. As shown in Fig. 7, gaze labels form a smooth spherical manifold. Isomap best preserves global spherical geometry, while LLE (Roweis & Saul, 2000) collapses global structure and t-SNE (Maaten & Hinton, 2008) distorts geometry and amplifies noise. To enable end-to-end training, we introduce a lightweight Isometric Propagator **(IP)**—a 3-layer MLP trained to approximate Isomap embeddings with high fidelity (L1 error 0.05, angular error < 0.3°) and is frozen during training. This preserves geometric structure while avoiding Isomap's computational cost.

## 5.5. Qualitative Results

**(1) VCR-Enhanced Gaze Distribution Diversity** As shown in Fig. 9(a), models without VCR produce clustered predictions biased toward dominant training viewpoints, severely limiting gaze coverage. In contrast, VCR enables a more diverse and uniform distribution across extreme angles [Fig. 9(b)], effectively bridging the gap between controlled

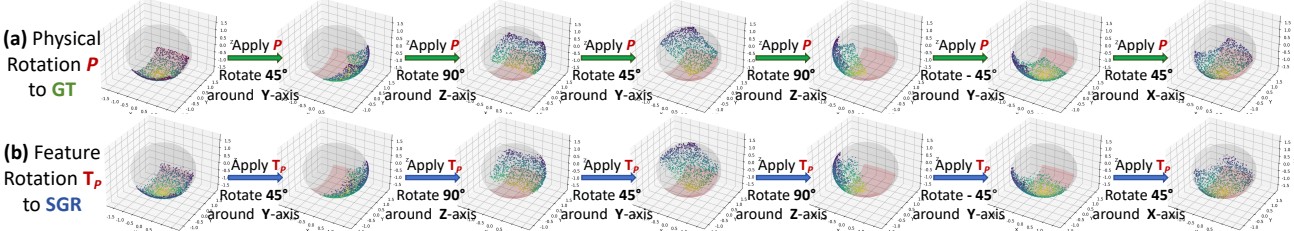

*Figure 8.* Feature-space rotation validation. (a) GT distributions under 3D physical rotations $P$ around X/Y/Z axes. (b) SGR transformed by $T_P$ and projected via Isomap maintain a high degree of consistency with the GT distribution after physical rotation on all rotation axes.

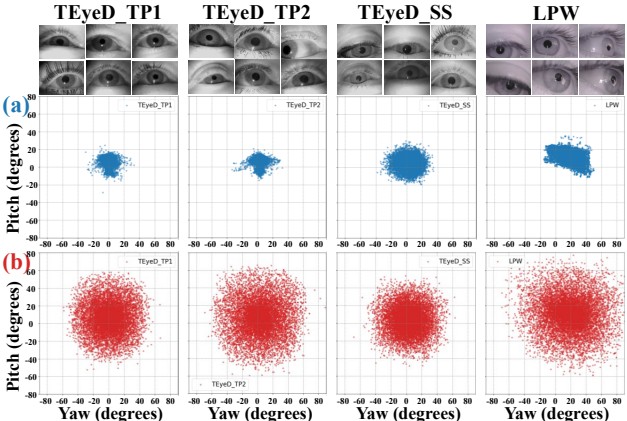

*Figure 9.* Distribution of gaze. (a) Without VCR, predictions concentrate within a narrow range. (b) With VCR, distributions become more diverse and spread, covering a larger field-of-view.

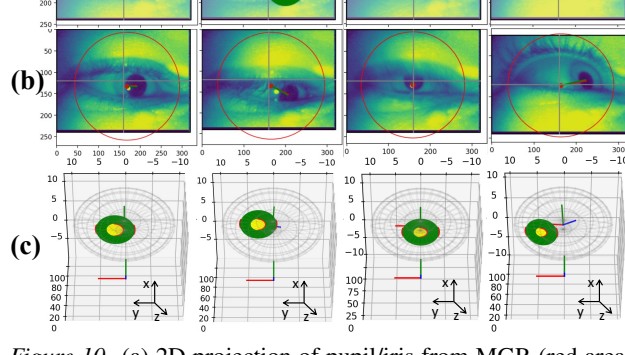

*Figure 10.* (a) 2D projection of pupil/iris from MGR (red areas represent the edge sampling points). (b) Predicted gaze (green) vs. ground truth (red). (c) Reconstructed 3D eyeball model.

and in-the-wild domains.

**(2) VCR-Enhanced Rotation Equivariance** We validate physical consistency by comparing ground-truth distributions under physical 3D rotations $P$ with SGR transformed by $T_P$ and projected via Isomap (Fig. 8). The strong alignment between the two, and quantitative results ($\epsilon_{\text{gaze}} < 0.96°$, $\epsilon_{\text{struct}} < 1.88\text{px}$) confirm that our feature-space rotation $T_P$ effectively approximates genuine 3D physical transformations $P$.

**(3) Prediction and Rendering Visualization** The 3D eye model and estimated gaze direction are projected onto the 2D image plane. As shown in Fig. 10, the predicted 3D gaze vectors [Fig. 10(b)] remain consistent with the viewing direction across frames, while the reconstructed 3D eye model [Fig. 10(c)] aligns well with pupil and iris projections [Fig. 10(a)], demonstrating accurate structural recovery and geometrically plausible 2D rendering.

## 6. Discussion and Conclusion

**(1) Robustness to Low-Quality or Noisy Inputs** Although trained on near-eye images, SG-Gaze does not rely on high-resolution textures. When edge cues become unreliable, the dual-branch design provides complementary fallback: AGE supplies geometric consistency, MGR contributes structural priors, and inference requires no 2D semantics. Meanwhile,

VCR introduces blur-, scale-, and view-perturbations, collectively enabling robustness under low-resolution or degraded inputs. Extensive experiments confirm stable performance under varying quality, illumination, and segmentation noise.

**(2) Modeling Simplification and Trade-offs** SG-Gaze prioritizes subject-invariant geometric consistency over subject-specific anatomy (e.g., kappa angle), trading slight in-domain accuracy losses for stronger cross-domain robustness. Weak 2D edge supervision favors physically plausible 3D structure over pixel-level precision. While MGR adopts simplified eyeball and camera models, AGE and adversarial alignment alleviate model mismatches, and device-specific fine-tuning can further improve precision. Consistent gains across 12 transfer tasks indicate that SG-Gaze leverages geometric invariances rather than dataset-specific bias.

**Conclusion** This work presents SG-Gaze, a framework that embodies *physics-as-representation* for gaze perception, integrating spherical geometry, anatomical structure, and viewpoint equivariance into a unified model. By treating physical laws as the foundation of representation, our approach demonstrates how physical inductive biases enhance interpretability and cross-domain generalization. Beyond gaze estimation, this work advances a broader paradigm: that encoding structural, geometric, and equivariant priors can bridge appearance-driven perception with physically consistent understanding—moving toward machines that see not only pixels, but the underlying physical reality.

## Acknowledgements

This work was supported in part by the grants from the National Natural Science Foundation of China under Grant 62332019 and 62576356, the National Key Research and Development Program of China (2023YFF1203900, 2023YFF1203903), the Natural Science Foundation of Shandong Province (ZR2023QD087), China Postdoctoral Science Foundatation (2024M764316), Sponsored by Beijing Nova Program (20240484513).

## Impact Statement

This paper presents a representation learning framework that incorporates structural and geometric inductive biases for physically constrained prediction tasks, with 3D gaze estimation used as a concrete case study.

**Potential benefits.** The proposed approach contributes to the broader field of machine learning by demonstrating how physically grounded constraints and equivariant regularization can improve interpretability and cross-domain generalization of learned representations. Beyond gaze estimation, similar principles may be applicable to other domains involving geometric structure and physical consistency, such as pose estimation, robotics perception, or scientific data analysis. By promoting representations that are more interpretable and robust to domain shifts, this line of work may help reduce reliance on large amounts of domain-specific labeled data.

**Potential risks and limitations.** Gaze estimation technologies, including those that could benefit from improved representations, may raise privacy or surveillance concerns if deployed irresponsibly. However, this work does not introduce new sensing modalities, data collection procedures, or deployment mechanisms, and focuses solely on representation learning methods using existing datasets. The proposed framework does not infer personal attributes beyond gaze direction, nor does it enable identification or tracking of individuals.

Additionally, improved robustness and interpretability may lead to increased trust in learned models. As with other machine learning systems, over-reliance on model outputs without proper validation could result in misuse or misinterpretation in downstream applications.

**Outlook.** We encourage future research to consider the responsible deployment of physically grounded representation learning methods, particularly in human-centered applications. Further work could explore how such representations interact with privacy-preserving learning, user consent, and transparent system design. Overall, we view this work as advancing general principles of representation learning rather than introducing application-specific societal risks.

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
